# An Update on Applications of Cattle Mesenchymal Stromal Cells

**DOI:** 10.3390/ani12151956

**Published:** 2022-08-02

**Authors:** Barbara Merlo, Penelope Maria Gugole, Eleonora Iacono

**Affiliations:** 1Department of Veterinary Medical Sciences, University of Bologna, 40064 Ozzano Emilia, BO, Italy; penelopemaria.gugol2@unibo.it (P.M.G.); eleonora.iacono2@unibo.it (E.I.); 2Interdepartmental Center for Industrial Research in Health Sciences and Technologies, University of Bologna, 40126 Bologna, BO, Italy

**Keywords:** bovine, mesenchymal stromal cells, clinical applications, reproductive biotechnologies

## Abstract

**Simple Summary:**

Among livestock species, cattle are crucially important for the meat and milk production industry. Cows can be affected by different pathologies, such as mastitis, endometritis and lameness, which can negatively affect either food production or reproductive efficiency. The use of mesenchymal stromal cells (MSCs) is a valuable tool both in the treatment of various medical conditions and in the application of reproductive biotechnologies. This review provides an update on state-of-the-art applications of bovine MSCs to clinical treatments and reproductive biotechnologies.

**Abstract:**

Attention on mesenchymal stromal cells (MSCs) research has increased in the last decade mainly due to the promising results about their plasticity, self-renewal, differentiation potential, immune modulatory and anti-inflammatory properties that have made stem cell therapy more clinically attractive. Furthermore, MSCs can be easily isolated and expanded to be used for autologous or allogenic therapy following the administration of either freshly isolated or previously cryopreserved cells. The scientific literature on the use of stromal cells in the treatment of several animal health conditions is currently available. Although MSCs are not as widely used for clinical treatments in cows as for companion and sport animals, they have the potential to be employed to improve productivity in the cattle industry. This review provides an update on state-of-the-art applications of bovine MSCs to clinical treatments and reproductive biotechnologies.

## 1. Introduction

Research into stem cells has been very active over the past decade. Due to the increasing number of studies, several breakthroughs have been achieved in this field, and stem cell therapy has gained ground as a modality of regenerative medicine. Mesenchymal stromal cells (MSCs) are present in different body tissues and are characterised as able to adhere to plastic, express specific surface antigens and possess multipotent differentiation potential [1]. Furthermore, they are good candidates for the treatment of various diseases due to characteristics such as low immunogenicity, anti-inflammatory potential and their ability to produce various mediators and molecules that help the regenerative function [2].

Bovine MCSs have been isolated and characterised (Figure 1) from different adult and foetal tissues, including bone marrow (BM) [3,4,5,6,7,8,9,10,11,12,13,14,15,16,17,18,19,20,21,22,23,24,25,26,27,28,29,30,31,32,33,34,35,36,37,38], endometrium (EN) [39,40,41,42,43,44,45,46,47,48], adipose tissue (AT) [29,30,31,32,34,37,49,50,51,52,53,54,55,56,57,58,59,60,61,62,63,64,65,66] and foetal liquid and adnexa, such as umbilical cord blood (UCB) [67,68,69,70], Wharton’s jelly (WJ) [58,71,72,73], umbilical cord matrix (UC) [74,75,76,77,78,79,80,81,82], amnion (AM) [83,84], amniotic fluid (AF) [57,83,85,86,87,88] and placenta (PL) [37,89,90]. Less common sources of bovine MSCs have been foetal liver [91], dermal tissue [92], foetal lung tissue [93], embryo yolk sack [94], synovial fluid [95], milk [96], pericardium membrane [97], pancreas [98], tongue epithelium [99], skeletal muscle [65,100] and peripheral blood [48,101].

The potential of MSCs for cell-based therapies has originally been based on their typical characteristics, which include the multipotentiality to differentiate in vitro into mesodermal-derived lineages, particularly osteogenic, chondrogenic and adipogenic cells [1]. Furthermore, it has been demonstrated that the paracrine activity of MSCs exerts therapeutical effects involving regeneration, immunomodulation, angiogenesis and antiapoptosis [102,103,104].

The immunomodulatory activity of MSCs depends on direct cell-to-cell contact and on contact-independent paracrine signalling, with the production of soluble factors regulating proliferation, differentiation, migration and apoptosis of several immune cells [105]. The reduced immunogenicity of MSCs is another aspect that strengthens their potential for cell therapy related in part to the low expression of major histocompatibility complexes I and II (MHC-I and II) and to the absence of expression of T-cell costimulatory molecules (CD40, CD80 and CD86) [106]. Taking together the immune regulatory abilities and reduced immunogenicity, allogeneic MSCs transplanted into recipients are able to escape direct recognition by natural killer cells and prevent activation of T lymphocytes, possibly also reducing the potential activation of the indirect pathway by the presentation of donor-derived MHC-I/II peptides by antigen-presenting cells to B cells and subsequent alloantibodies production [107]. Therefore, low immunogenicity may result in higher efficacy and lower risk of local inflammation following MSCs administration, reducing potential adverse effects [107]. In cattle, it has been demonstrated that foetal AT-MSCs and BM-MSCs respond to inflammatory stimulation with interferon γ (IFNγ) by increasing immune-related gene expression and activity in a dose-dependent manner and upregulating gene expression of IL-6 [30]. However, conditioned medium from IFNγ-stimulated and unstimulated BM-MSCs and AT-MSCs exerts similar suppression of proliferation of alloantigen-activated bovine peripheral blood lymphocytes [30]. Whereas immunomodulatory properties appear to be similar between BM-MSCs and AT-MSCs, higher expression of MHC-I and MHC-II in BM-MSCs suggested that the immunogenic potential of bovine foetal MSCs might be tissue-dependent and that AT-MSCs might be more suitable candidates for allogeneic therapy [30].

Autologous MSCs therapy implies cell isolation and expansion to achieve therapeutic doses. Consequently, there is a lag time between their collection and use, threatening the effectiveness of the treatment. In addition, critical parameters for MSCs isolation include donor variability, tissue of origin, amount of tissue and culture conditions [108]. On the other hand, foetal- and placental-derived MSCs have been found superior to adult MSCs as candidates for allogeneic therapeutic applications due to their lower immunogenicity [109,110]. Cryopreservation represents an efficient method for the preservation and pooling of MSCs to obtain the cell counts required for clinical applications. Samples can be harvested, and then cells can be isolated, expanded and stored for later use, optimising logistics from collection to transplantation. Accordingly, the ability of MSCs to survive long periods of storage and, at the same time, maintain their qualities is critical for the development of allogeneic cell therapies. Upon cryopreservation, it is important to preserve MSCs’ functional properties, including immunomodulatory properties and multilineage differentiation ability. Further, a biosafety evaluation of cryopreserved MSCs is essential prior to their clinical applications [111]. Considering cattle, Oyarzo et al. compared PL-MSCs and foetal MSCs originated from AT and BM in order to assess their ability to survive different cryoprotectant solutions exposure [37]. While the apoptotic potential was similar, foetal AT-MSCs and PL-MSCs presented consistently higher percentages of viability than did foetal BM-MSCs [37]. On the other hand, AT-MSCs were more resistant than PL-MSCs, but the latter have the advantage of coming from a readily available tissue usually considered waste, without ethical concerns [37].

Although in veterinary medicine, cell therapies are mainly focused on pets, regenerative medicine applications also involve farm animals, not only for their importance as a food source [112] but also as models [113]. Among livestock species, cows have a high economic impact, and reproductive biotechnologies are routinely applied [114,115]. The dairy and beef industries are essential for food production. Dairy products and ruminant meat provide essential elements for the human diet. According to the Food and Agriculture Organization (FAO), there are almost 1.5 billion cattle in the world. Cows produce 81 per cent of global milk production, and the world demand for beef is projected to increase to 75 million tonnes by 2030 [116]. Animal health is an important issue related not only to animal welfare itself but also to the One Health perspective, in which human, animal, plant and environmental health are interdependent. This review summarises the applications of MSCs in cattle to treat clinical conditions and improve reproductive biotechnologies.

## 2. Bovine MSCs for Clinical Treatments

So far, MSCs have been used in many experimental instances to treat various diseases in different animal species. Orthopaedic diseases were the primary field of regenerative veterinary medicine, and then the focus rapidly expanded to other areas. Dogs and horses were the species in which stem cell-based therapies were commonly used to treat different diseases of various organ systems, while for cats, they were used for renal, respiratory and inflammatory pathologies [117]. Bovine MSCs can be potentially used in various clinical conditions. Nevertheless, the application of novel MSCs therapies in large ruminants is still limited.

The major obstacles in livestock species are related to a minor interest in treating clinical conditions in these animals compared to pets and the higher maintenance costs in comparison with other animal models [118]. Laboratory animals or small animals are usually preferred as models to start any research for human pathologies due to the reasonable buying and care costs together with easier manageability and housing. However, for a better understanding and a thorough evaluation of cell-based therapies, various animal models are necessary to successfully move from the laboratory bench to human health applications. The development of products for animal use has the advantage that they can be immediately tested in the target species. This aspect not only allows to understand the potential of MSC-based products for clinical application in animals but may also provide models for similar human applications [119].

Although many studies have been published for animal MSCs, it is still not easy to evaluate the efficacy of MSC-based therapies because of the different sources of MSCs and variations in manufacturing processes, inconsistent characterisation and measure of potency, inappropriate controls and a lack of experimental power [119]. MSCs have been isolated from different sources and, depending on the tissue of origin, they may possess different properties, which should be taken into account when choosing the optimal stem cell therapy for a specific pathology in order to achieve successful results. On the other hand, there is no evidence for a favoured tissue as an MSCs source due to the presence of a wide variability between donors [108,120].

### 2.1. Chronic Wound Healing

In the last years, the application of regenerative medicine to skin lesions has been a focus for both human and veterinary medicine. The physiological healing process of cutaneous wounds is a well-orchestrated complex of molecular and biological activities. Even so, a chronic lesion can develop when the normal process fails. The regenerative potential of stromal cells has also been widely recognised for skin lesion repair [121]. Recent studies support the concept that MSCs can be appropriated for treating chronic wounds [122,123,124].

Even if the exact functions of stromal cells in wound healing have not yet been completely elucidated, they are involved in the removal of dead cells and necrotic tissue, angiogenesis, reduction in scar tissue formation, contraction of the wound and induction of re-epithelisation [121]. Consequently, wound healing is promoted, and local inflammation is reduced. Table 1 summarises the studies regarding MSC applications for wound healing in cattle.

The first report of a case study in which autologous BM-derived MSCs were used to treat a chronic ulcer in a heifer dates back to 2012 [10]. A 2-year-old Jersey heifer had been suffering from a chronic nonhealing ulcerative wound involving full-thickness skin and underlying muscle in the lumbar region for 4 months. Standard therapies were ineffective, so a clinical trial was made with autologous BM-MSCs. Bone marrow was aspirated from the tibia, and MSCs were isolated, expanded and then diluted in saline solution for intradermal and topical implantation in the wound. Various parameters and measures were monitored during the trial. At histopathology, the progression of the healing process was observed since neovascularisation appeared, as well as fibroblasts, sebaceous glands and epithelialisation. The content of collagen was increased after stem cell therapy, and the healed tissue was progressing towards physiological stretchability and tensile strength. The 4-month-old chronic wound healed within 18 days, indicating that MSCs application could be an effective therapeutic approach for nonhealing chronic wounds [10].

Another clinical study of the same research group concerns the successful treatment of an interdigital chronic ulcerative wound in a 6-year-old cross-bred Jersey cow [13]. The animal presented with a 4-month interdigital hoof lesion nonresponding to conventional treatments. Autologous BM-MSCs therapy was also used for this patient. Granulated tissue rapidly grew, and the healing process was completed in 18 days. The parameters analysed to assess the progression of the healing process confirmed the clinical process, and the pain-free walking distance evaluation was gradually increased over the study period [13].

In the last clinical trial [14], a bull was presenting a wound in a hind limb above the hock joint as a consequence of a car accident, which had happened 8 months before. Different local treatments and antibiotic courses turned out as unsuccessful as chemical and cryocauterisation. Autologous BM-MSCs application was performed. Similar protocols were used for the collection, isolation and expansion of BM-MSCs, but in this case, some cells were intravenously administered in addition to local treatment. Healing was completed within 4 weeks, and the evaluated parameters confirmed the outcome [14].

Despite the lack of controls and large-scale randomised studies and clinical trials, the promising results obtained from the applications of autologous BM-MSCs confirmed the potential of MSC-based therapy for treating chronic nonhealing wounds in bovines.

A weak immunogenicity and a vasculogenic effect are favourable properties for wound healing capacity. Bovine BM-MSCs are the most well-characterised cells, and recently their immunomodulatory properties [30] and proangiogenic potential [31] have been investigated. Comparing bovine foetal MSCs derived from bone marrow and adipose tissue, both upregulated the expression of immunomodulatory genes and showed similar in vitro immunomodulatory ability, while the lower expression of MHC-I and MHC-II suggested that AT-MSCs might be less immunogenic compared with BM-MSCs [30]. Furthermore, BM-MSCs displayed similar migratory ability, higher proliferative capacity and lower proangiogenic potential compared with AT-MSCs [31]. These results might suggest that bovine AT-MSCs could be even more promising than BM-MSCs in enhancing the treatment of chronic wound healing.

### 2.2. Mastitis

In the dairy industry, mastitis is a common problem, which implicates costs to treat the disease, and since antimicrobials are the standard therapy, this increases the possibility of developing antimicrobial resistance. Hence, alternative therapies are required.

The mammary gland contains stromal cells and precursors with high regenerative potential, which apparently are maintained during the productive life of dairy cows. The presence of such cells opens new research perspectives regarding the physiological mechanisms concerned with milk secretion and the possibility of enhancing or prolonging dairy cow production [125]. The presence of a subpopulation of adult stromal cells in the mammary gland was first demonstrated in human and mouse [126,127]. Then, in the cow, three different colony morphologies were isolated, suggesting the existence of different progenitor populations and of an epithelial cell hierarchy in the bovine mammary gland similar to humans [128]. Such stromal/progenitor cells have been largely investigated [125,129].

On the other hand, less research is available for MSCs and bovine mammary glands. As summarised in Table 2, different in vitro studies showed that UC-MSCs could promote milk protein and fat synthesis and the expression of key genes in bovine mammary gland epithelial cells via IGF-1 [75,76,78] and reduce their apoptosis rate [77]. Furthermore, it has been demonstrated that bovine MSCs have antibacterial activity [29]. The conditioned medium from bovine foetal MSCs obtained from bone marrow and adipose tissue showed in vitro antibacterial potential against *S. aureus*, a mastitis-causing pathogen, by reducing about 30% of relative bacterial growth [29]. The mechanisms that regulate the antibacterial activity of bovine MSCs have not been totally elucidated, but the expression of β–defensin 4 A and NK-lysine 1, two antibacterial peptides, was associated with the in vitro effect of such MSCs [29].

Dairy cows were experimentally infected to induce *S. aureus* clinical mastitis in order to evaluate the safety and efficacy of an allogenic MSC-based therapy [62]. Bovine foetal AT-MSCs were intramammary inoculated twice (days 1 and 10) during a 20-day experimental period. No clinical or immunological response was induced in healthy cows, and the bacterial count in milk was reduced in MSC-treated cows compared with controls [62]. A similar decrease in somatic cell count (SCC) in the milk of mastitic animals was observed in cows treated intramammary with a single administration of allogenic AT-MSCs during a 15-day experiment [64]. On days 3 and 7, maximum expression of anti-inflammatory cytokines (IL-6, IL-10), antimicrobial peptides (cathelicidin, lipocalin and cystatin) and angiogenic genes (angiopoietin) was observed [64]. With the aim of preventing subclinical mastitis, UCB-MSCs and extracellular vesicles (EVs) were injected locally and IV on days 0 and 7 in healthy (safety trial) and subclinical mastitis cows [68,69]. Both MSCs and EVs were safe, and all treated cows were cured permanently within 15 days [68]. Treated animals showed a reduced SCC in mastitic milk compared with the control (antibiotic) group, an enhancement in the expression of anti-inflammatory cytokines, antimicrobial peptides and angiogenic genes and a decrease in the expression of proinflammatory cytokines [68,69]. Finally, a conditioned medium from bovine AM-MSCs (2 h coincubation in phosphate-buffered saline (PBS)) was used to treat mastitis in comparison with conventional antibiotics [84]. Milk pH value and titratable acidity were similar between treatments, while the level of ionic calcium concentration decreased 3 days later in MSCs-treated cows compared with antibiotic-treated animals [84]. Moreover, the somatic cell number was similar in both groups, demonstrating that conditioned medium from bovine AM-MSCs has the therapeutic potential to treat bovine mastitis and might replace antibiotics in the future [84].

### 2.3. Reproductive System

In the last 50 years, the selection in the dairy industry has led to an improvement in average milk production by a single cow. However, the selection for milk yield has caused some unfavourable effects, such as a decrease in fertility. Despite an improvement in cow fertility in the last two decades, as a consequence of selection for fertility traits in breeding programmes and improvement in animal nutrition and comfort, reproductive performance is not optimal yet [130]. Reproductive disorders are directly correlated with low fertility in dairy cows.

The endometrium is characterised by an elevated and constant regeneration, and mesenchymal progenitor cells have also been identified in the cow endometrium [39]. Progenitor cells were isolated and characterised in cyclic cows [40,42,44] and heifers [41] and were able to respond after challenging with lipopolysaccharide (LPS) [43]. Furthermore, the presence of endometrial MSCs was also confirmed in the postpartum period in both healthy cows and those affected by endometritis [45]. In this period, uterine involution occurs, involving endometrial regeneration [131], and the presence of pathogenic bacteria needs to be controlled in the uterus for fertility restoration. However, pathogenic bacteria are not always rapidly eliminated and often generate uterine disease (metritis and endometritis), leading to reduced fertility [132]. Endometrial MSCs from bovine inflamed uteri showed modified characteristics, especially in clinical than in subclinical endometritis, and the in vitro exposure of endometrial MSCs to PGE2, a mediator of inflammation, modified their transcriptomic profile [45]. Bovine endometrial MSCs have also been immortalised from lines derived in different phases of the oestrous cycle [47]. Immortalised cells maintained mesenchymal and immunomodulatory characteristics, with an increased migratory capacity towards an inflammatory niche but a decreased answer to embryonic cytokine expression at implantation [47]. Interestingly, combined proinflammatory and implantation signals ensured the retention of endometrial MSCs in case of pregnancy, while they showed a mesenchymal to epithelial transition state in the absence of an embryo [47]. Despite research into bovine endometrial MSCs, no report exists about their application in treating cow uterine inflammations. On the other hand, bovine MSCs derived from adipose tissue showed an inhibitory effect on in vitro LPS challenge of endometrial epithelial cells [63]. When used in vivo to treat metritis, allogenic AT-MSCs did not induce any immunological rejection response in treated animals (IV, local, IV + local), and all cows were completely and permanently cured within 30 days after treatment [64]. Polymorphonuclear (PMN) cell count was reduced in cervical vaginal fluid and the expression of IL-6, IL-10, cathelicidin, lipocalin, cystatin and angiopoietin were observed at day 3 in the IV + local group [64]. More recently, UCB-MSCs and their EVs have also been successfully used for metritis treatment by the same research group [70]. Moreover, in this case, a higher decrease in PMN was observed for MSCs and EV-treated cows compared with antibiotic-treated ones, as well as an increase in the expression of anti-inflammatory cytokines [70].

Other pathologies, which can lead to considerable economic loss, are those involving the ovaries. Ovarian dysfunctions in dairy cattle have a high incidence and are responsible for a reduction in reproductive performance. The two major ovarian causes of infertility in dairy cows are inactive ovaries and ovarian cysts [133,134]. Chang et al. transplanted AF-MSCs into cows affected by bilateral ovarian dystrophy in an attempt to restore or improve ovarian function [87]. Each ovary was injected with 50 μL of PBS containing 0.58 million cells, and then cows were monitored for oestrus and inseminated [87]. Half (4/8) of the animals treated with AF-MSCs showed oestrus, and two of them delivered a calf, while no oestrus was observed in control animals, demonstrating that MSCs therapy is a potentially useful treatment to alleviate the impact of ovarian dystrophy in dairy cows [87]. Peng et al. injected PL-MSCs into ovarian cysts with or without fluid drainage and compared them to control animals and GnRH-treated animals [90]. The use of PL-MSCs allowed for recovery and conception [90], indicating a new therapeutic potential of these cells and a possible alternative to hormones in the treatment of cattle ovarian cysts. Finally, the intraovarian injection of MSCs was used to reduce the negative effects of repeated ovum pick-up (OPU) under acute and chronic scenarios in bovines [61]. In fact, this technique is generally considered a safe way to collect oocytes from live donors but inevitably causes trauma to the ovarian tissue, and repeated procedures over years are associated with a progressive decrease in oocyte yield [61]. For the experiment, one ovary was injected with 2.5 million AT-MSCs, and the other one was used as the control [61]. MSCs had beneficial effects on the fertility of acute OPU injured cows, but not in cows with chronic ovarian lesions [61]. In this case, it was speculated that MSCs could no longer restore the compromised follicular population or ovarian physiology in cows with chronic inflammatory processes in the ovaries due to repeated OPU over time [61]. The overall MSC clinical applications for the reproductive system are presented in Table 3. 

## 3. Bovine MSCs for Reproductive Biotechnologies

The first successful nuclear transfer (NT) dates back to 1952, when the nucleus from an early tadpole embryo was transferred into an enucleated frog egg [135]. Then, in 1996, Dolly was the first mammalian cloned using an adult somatic cell as a nucleus donor [136]. Somatic cell nuclear transfer (SCNT) (Figure 2) is an important research tool since it permits a differentiated cell to be reprogrammed to a totipotent state [137]. The donor cell is a key factor in the process, and interest in bovine SCNT led to consider MSCs as appropriate candidates due to their characteristics. Studies using bovine MSCs from different sources for NT were carried out and are summarised in Table 4. 

Firstly, it was demonstrated that bovine BM-MSCs had developmental totipotency after NT [3] and were better than adult fibroblasts in driving the preimplantation development of cloned embryos efficiently [5]. In another study investigating the epigenetic status of donor cells to improve SCNT [52], it was demonstrated that bovine AT-MSCs at passage 5 had the highest level of multipotency and the lowest level of chromatin compaction. Bovine AF and AT-MSCSs were then successfully used to produce embryos and calves after NT [53], and in vitro development of bovine embryos cloned using less methylated AF and AT-MSCS was improved using trichostatin A [57]. Pregnancies were also obtained after the transfer of blastocysts derived from WJ-MSCs NT [72]. A higher potential for AM and AT-MSCs than adult fibroblasts was observed in terms of blastocysts obtained after oocyte reconstruction [138]. More recently, epigenetic reprogramming events were investigated, and it was observed that the SCNT embryos derived from bovine AT-MSCs endured considerable nuclear reprogramming during early embryo development [56]. Finally, in an attempt to improve NT efficiency, the aggregation of two AT-MSC-derived embryos seemed to positively affect embryo quality, which may improve postimplantation development [66]. 

Another context of research into cells includes their ability to incorporate exogenous DNA for the production of transgenic animals. Bovine MSCs were transfected with pBC1-anti-CD3 vector, and while those derived from WJ were more sensitive to treatments, AT-MSCs showed a better response to transfection [58].

Bovine MSCs have also been used for in vitro embryo production. The traditional coculture system of bovine embryos with granulosa cells was less efficient than coculture with AT-MSCs [54]. In addition to increasing blastocyst rates, MSCs coculture also improved embryo quality, with an increase in total cell numbers and mRNA expression levels for POU5F1 and G6PDH [54]. It was speculated that the paracrine capacity of MSCs could be responsible for the positive effects observed [54].

Another application of MSCs is to produce germinal cells after differentiation. The in vitro production of germ cell lineages is a new intriguing strategy for obtaining gametes in order to treat infertility, disseminate the genetics of elite animals and preserve endangered species [139]. The in vitro effect of bone morphogenetic protein 4, transforming growth factor β1 and retinoic acid on the potential for germ cell differentiation of bovine foetal BM-MSCs was investigated [27]. The stimulated cells expressed pluripotent markers OCT4, NANOG and male germ cell gene DAZL, demonstrating their potential for early germ cell differentiation [27]. When coculturing bovine foetal BM and AT-MSCS with Sertoli cells, cell morphology modifications were induced, as well as variations in the expression profiles of mesenchymal, pluripotent and germ cell genes, suggesting progression of AT-MSC into early stages of germ cell differentiation and advancement of BM-MSCs into the multipotent state [34].

## 4. Conclusions

The development of stem cell technologies in species other than bovine can be seen as a useful background for developing and deepening similar advancements in livestock. MSC characteristics make them appealing for their potential in clinical applications, and the lack of ethical concern is the other factor that makes them ideal for laboratory studies. As for humans [140], for successful cell-based therapies, stem cells must be able to differentiate into specific targeting cells or must act via paracrine mechanisms. Their extraction and isolation must be feasible, and transplantation must be effective and safe. Furthermore, ex vivo cell expansion is required since a considerable number of cells is essential to optimise the therapeutic effects. However, the lifespan of MSCs is limited during in vitro culture, and their senescence is a limit from the viewpoint of clinical applications. On the one hand, the limited cell proliferation potency protects them from malignant transformation after transplantation; on the other, senescence can alter various cell functions essential for therapeutic efficacy, such as proliferation, differentiation and migration. Therefore, after in vitro expansion and before therapeutic use, it should be considered whether these cells still possess stemness properties.

The bovine model could be advantageous for the size and physiology when compared with traditional laboratory animals. In cattle, MSCs have been isolated from different tissues, and their pluripotency has been demonstrated, but there is still a lack of clinical applications and studies comparing MSCs from different sources to suggest which one is the best choice for cell therapy or for which specific pathology. The studies presented are promising for the possible applications of MSCs both in veterinary medicine and the livestock industry. However, more studies are required to develop bovine-specific protocols, and further investigation is needed to evaluate clinical responses after cell therapy applications. Attitudes in the livestock industry have shifted towards the preservation of the commercial viability of individual animals with high genetic value, leading, in turn, to an increase in medical expenditure to keep those animals healthy [141]. MSCs treatment has the potential to reduce animal recovery time and reduce economic loss associated with bone and joint injury, reducing the time for repair that can negatively influence milk and meat production and interfere with natural breeding [141]. Nevertheless, orthopaedic applications have not yet been applied clinically in cows. The antimicrobial activity of MSCs and their derivatives has great potential for the treatment of conditions such as mastitis. In addition to the direct impact on milk production in the dairy industry, it would provide an alternative to the use of antimicrobials, reducing the possibility of antimicrobial resistance and the presence of antibiotics in milk. MSCs treatment has the potential to decrease recovery from various diseases affecting production, thus increasing profitability.

## Figures and Tables

**Figure 1 animals-12-01956-f001:**
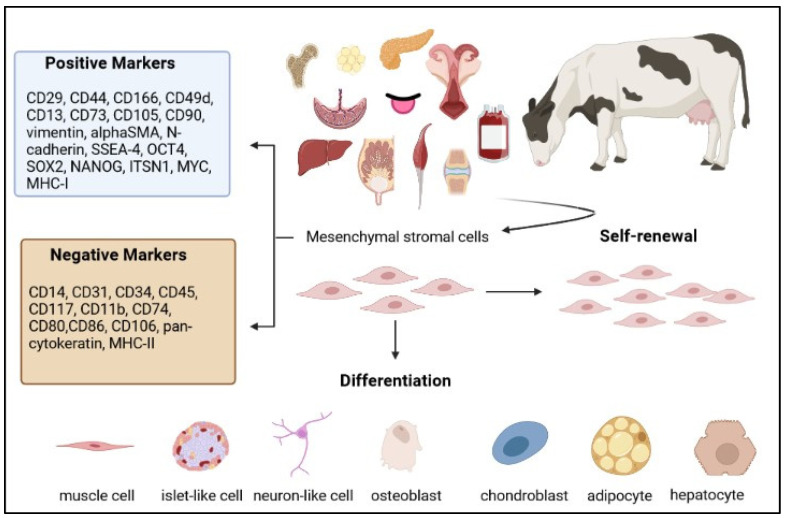
Schematic diagram of the characteristics of bovine mesenchymal stromal cells (created in Biorender.com, accessed on 14 July 2022).

**Figure 2 animals-12-01956-f002:**
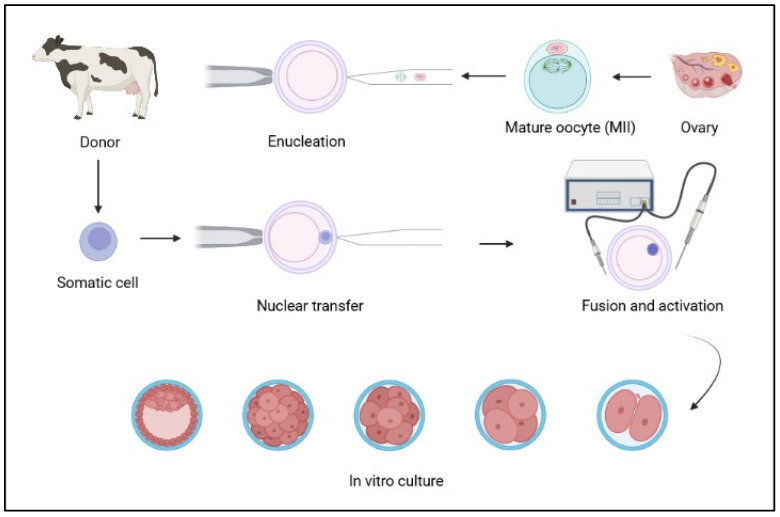
Schematic diagram of the somatic cell nuclear transfer (SCNT) technology (created in Biorender.com, accessed on 14 July 2022).

**Table 1 animals-12-01956-t001:** Bovine MSC applications for wound healing.

Source	Application	References
Bone marrow	Autologous treatment of a chronic ulcer in a heifer	[10]
Bone marrow	Autologous treatment of an interdigital chronic ulcerative wound in a cow	[13]
Bone marrow	Autologous treatment of a wound in a hind limb of a bull	[14]

**Table 2 animals-12-01956-t002:** Bovine MSC applications for the mammary gland.

Source	Application	References
Umbilical cord	In vitro effects on mammary gland epithelial cells	[75,76,77,78]
Bone marrow, adipose tissue	In vitro effects on *S. aureus*	[29]
Adipose tissue	In vivo effects on *S. aureus*-induced mastitis	[62]
Adipose tissue	In vivo effects on mastitis	[64]
Umbilical cord blood	In vivo effects on subclinical mastitis	[68,69]
Amniotic membrane	In vivo effects of conditioned medium to treat mastitis	[84]

**Table 3 animals-12-01956-t003:** Bovine MSCs from different sources for treatment of reproductive system diseases.

Source	Application	References
Adipose tissue	Metritis	[64]
Umbilical cord blood	Metritis	[70]
Amniotic fluid	Bilateral ovarian dystrophy	[87]
Placenta	Ovarian cysts	[90]
Adipose tissue	Intraovarian injection for repeated OPU lesions	[61]

**Table 4 animals-12-01956-t004:** Bovine MSCs from different sources as nucleus donors for nuclear transfer.

Source	References
Bone marrow	[3,5]
Adipose tissue	[52,53,56,57,66,138]
Amniotic fluid	[53,57]
Amniotic membrane	[138]
Wharton’s jelly	[72]

## Data Availability

No new data were created or analysed in this study. Data sharing is not applicable to this article.

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
