# Peer review of "An Update on Applications of Cattle Mesenchymal Stromal Cells"

_animals, 2022, doi:10.3390/ani12151956_

Round 1

Reviewer 1 Report

There are some limitations in explaining the terms used in this article. Although the information provided in the manuscript is more informative, the grammar used in the article is making it difficult to understand, that’s why it needs full proofreading to bring clarity to the meaning of how the experiments had been carried out. Since it’s a review article and the ideas mentioned are already utilized by different researchers providing more figures data and tables will make it easy for the readers to understand quickly and easily. In the article, the journal's guidelines were properly followed in terms of references and explaining the tables. The following are some of the questions and changes that need to be addressed.

In line 197 you mentioned IL-6, IL-10, Canthelicidin, and other different genes observed. These genes have nothing to do with milk production. How can you co-relate them with milk production parameters?

Q1. In the article several types of MSCs have been used so among them which one is the best to use?

Q2. In terms of economic costs, is it going to be affordable for average cow farm owners to use such MSCs for the treatment of various kinds of infections and diseases? 

Author Response

Comments and Suggestions for Authors

There are some limitations in explaining the terms used in this article. Although the information provided in the manuscript is more informative, the grammar used in the article is making it difficult to understand, that’s why it needs full proofreading to bring clarity to the meaning of how the experiments had been carried out. Since it’s a review article and the ideas mentioned are already utilized by different researchers providing more figures data and tables will make it easy for the readers to understand quickly and easily. In the article, the journal's guidelines were properly followed in terms of references and explaining the tables. The following are some of the questions and changes that need to be addressed.

 The manuscript was revised as suggested. Figures and data tables were added.

In line 197 you mentioned IL-6, IL-10, Canthelicidin, and other different genes observed. These genes have nothing to do with milk production. How can you co-relate them with milk production parameters?

“On day 3 and 7, maximum expression of anti-inflammatory cytokines (IL-6, IL-10), anti-microbial peptides (cathelicidin, lipocalin, and cystatin), and angiogenic genes (angiopoietin) were observed.”

Q1. In the article several types of MSCs have been used so among them which one is the best to use?

“In the bovine, MSCs have been isolated from different tissues and their pluripotency has been demonstrated, but there is still lack of clinical applications and studies comparing MSCs from different sources to suggest which one is the best for cell therapy or for which specific disease.”

Q2. In terms of economic costs, is it going to be affordable for average cow farm owners to use such MSCs for the treatment of various kinds of infections and diseases? 

“Attitudes in the livestock industry have shifted towards the preservation of the commercial viability of individual animals with high genetic value, leading in turn to an increase in medical expenditure to keep those animals healthy [140]. MSC treatment has the potential to reduce animal recovery time and reduce economic loss associated with bone and joint injury, reducing the time for repair that can negatively influence milk and meat production and interfere with natural breeding [140]. Nevertheless, orthopedic applications has not yet been applied clinically in cows. The antimicrobial activity of MSCs and their derivatives has a great potential for the treatment of conditions such as mastitis. In addition to the direct impact on milk production in the dairy industry, it would provide an alternative to the use of antimicrobials reducing the possibility of antimicrobial resistance and the presence of antibiotics in milk. MSCs treatment has the potential to decrease the recovery from various diseases affecting production, thus increasing profitability.”

Reviewer 2 Report

The authors of the review provided updates on the state of the art of the applications of bovine mesenchymal stem cells to clinical treatments and reproductive biotechnologies.

Overall, this is a clear, concise, and well-written manuscript. However some improvements should be made.

Minor issues:

- The possibility of using fresh or frozen cells is reported in the abstract but the topic is not taken up in the text of the manuscript; information relating to this topic could be reported.

-The introduction should be improved, the first two sentences (lines 56-58) should be rephrased and inserted at the end of the introduction and should be better argued.

-Lines 93-94: the sentence "Laboratory animals or small animals are usually preferred as models for human pathologies to start any research" should be better argued, motivating why laboratory animals are preferred: ethical, regulatory reasons, manageability, etc.

-Lines 256 and 263: the sentences could be rephrased and replaced "in a study" and "in another study" with the names of the authors of the reported studies.

-Double check the text and replace “µl” with “µL”.

Author Response

Comments and Suggestions for Authors

The authors of the review provided updates on the state of the art of the applications of bovine mesenchymal stem cells to clinical treatments and reproductive biotechnologies.

Overall, this is a clear, concise, and well-written manuscript. However some improvements should be made.

Minor issues:

- The possibility of using fresh or frozen cells is reported in the abstract but the topic is not taken up in the text of the manuscript; information relating to this topic could be reported.

“Autologous MSCs therapy implies cells isolation and expansion to achieve therapeutic doses. Consequently, there is a lag time between their collection and use, jeopardizing the effectiveness of the treatment. In addition, critical parameters for MSCs isolation include donor variability, tissue of origin, amount of tissue, and culture conditions [108]. On the other hand, fetal and placental derived MSCs have been found superior than adult MSCs as candidates for allogeneic therapeutic applications, due to their lower immunogenicity [109,110]. Cryopreservation represents an efficient method for the preservation and pooling of MSCs, to obtain the cell counts required for clinical applications. Samples can be harvested, expanded, and stored for later use, optimizing logistics from collection to transplantation. Accordingly, the ability of MSCs to survive long periods of storage and at the same time maintain their qualities is critical for the development of allogeneic cell therapies. Upon cryopreservation, it is important to preserve MSCs functional properties including immunomodulatory proper-ties and multilineage differentiation ability. Further, a biosafety evaluation of cryopreserved MSCs is essential prior to their clinical applications [111]. Considering cattle, Oyarzo et al. compared PL-MSCs and fetal MSCs originated from AT and BM in order to assesses their ability to survive to different cryoprotectant solutions exposure [37]. While the apoptotic potential was similar, fetal AT-MSCs and PL-MSCs presented consistently higher percentages of viability than fetal BM-MSCs [37]. On the other hand, AT-MSCs were more resistant than PL-MSCs, but the latter have the advantage of coming from a readily available tissue usually considered waste, without ethical concerns [37].”

-The introduction should be improved, the first two sentences (lines 56-58) should be rephrased and inserted at the end of the introduction and should be better argued.

Introduction was modified as suggested.

“Although in veterinary medicine cell therapies are mainly focused on pets, regenerative medicine applications involved also farm animals, not only for their importance as a food source [112], but also as models [113]. Among livestock species, cows have a high economic impact and reproductive biotechnologies are routinely applied [114,115]. The dairy and beef industries are essential for food production. Dairy products and ruminant meat provide essential elements for human diet. According to the Food and Agriculture Organization (FAO), there are almost 1.5 billion cattle in the world. Cows produce 81 percent of global milk production and the world demand for beef is projected to increase to 75 million tonnes by 2030 [116]. Animal health is an important issue related not only to animal welfare itself but also in the One Health perspective, in which human, animal, plant, and environmental health are interdependent. This review summarizes the applications of MSCs in cattle to treat clinical conditions and to improve reproductive biotechnologies”

-Lines 93-94: the sentence "Laboratory animals or small animals are usually preferred as models for human pathologies to start any research" should be better argued, motivating why laboratory animals are preferred: ethical, regulatory reasons, manageability, etc.

“Laboratory animals or small animals are usually preferred as models for human pathologies to start any research, due to the reasonable buying and care costs together with easier manageability and housing.”

-Lines 256 and 263: the sentences could be rephrased and replaced "in a study" and "in another study" with the names of the authors of the reported studies.

“Chang et al. transplanted AF-MSCs into cow affected by bi-lateral ovarian dystrophy in the attempt to restore or improve ovarian function [87].” “Peng et al. injected PL-MSCs into ovarian cysts with or without fluid drainage and compared to control animals and GnRH-treated animals [90].”

-Double check the text and replace “µl” with “µL”.

“…with 50 μL of PBS containing 0.58 million cells…”

Reviewer 3 Report

The present review summarizes the potential applications of bovine mesenchymal stromal cells with an emphasis on their clinical applications. The manuscript was very concise. However, many points should be considered as follows:
1. The title is not clear; the term bovine used in the title is a general term includes both cow and buffalo. However, the authors focused only on cow derived stem cells. the title should be modified to avoid misunderstanding. 

2. The introduction is short, more elaborations about the cow MSCs are missed. Moreover, robust statistic about the significance of cows in the meat and milk production industries are important.

3.  The immunomodulatory and anti-inflammatory roles of the cow MSCs should  be discussed.

4. The manuscript is missing the merits, and demerits of the use of cow MSCs in clinical field. 

5. Other clinical applications as orthopedic applications were not highlighted. IF these applications were not applied clinically, this should be addressed in the conclusion and future perspectives section.

6. More focus should be given on the future directions and possible future applications of cow MSCs.

7. The manuscript did not include any figures or illustrations that are important for the readers to understand the concept. 

Author Response

Comments and Suggestions for Authors

The present review summarizes the potential applications of bovine mesenchymal stromal cells with an emphasis on their clinical applications. The manuscript was very concise. However, many points should be considered as follows:
1. The title is not clear; the term bovine used in the title is a general term includes both cow and buffalo. However, the authors focused only on cow derived stem cells. the title should be modified to avoid misunderstanding. 

“An update on applications of cattle mesenchymal stromal cells”

  1. The introduction is short, more elaborations about the cow MSCs are missed. Moreover, robust statistic about the significance of cows in the meat and milk production industries are important.

The introduction was modified as suggested.

  1. The immunomodulatory and anti-inflammatory roles of the cow MSCs should  be discussed.

"The potential of MSCs for cell-based therapies has originally been based on their typical characteristics which include the multipotentiality to differentiate in vitro into mesodermal cell lineages including osteogenic, chondrogenic and adipogenic [1]. Furthermore, it has been demonstrated that the paracrine activity of MSCs exert therapeutical effects involving regeneration, immunomodulation, angiogenesis and anti-apoptosis [102–104]. The immunomodulatory activity of MSCs depend on direct cell-to-cell contact and contact independent paracrine signaling, with the production of soluble factors regulating proliferation, differentiation, migration and apoptosis of several immune cells [105]. The reduced immunogenicity of MSCs is another aspect that strengthen their potential for cell-therapy related in part to the low expression of major histocompatibility complexes I and II (MHC-I and II), and the absence of expression of T-cell costimulatory molecules (CD40, CD80 and CD86) [106]. Taking together immune regulatory abilities and reduced immunogenicity, allogeneic MSCs transplanted into recipients are able to elude direct recognition by natural killer cells and avert activation of T lymphocytes, possibly also reducing the potential activation of the indirect pathway by presentation of donor derived MHC-I/II peptides by antigen presenting cells to B cells and the subsequent allo-antibodies production [107]. Therefore, low immunogenicity may result in higher efficacy and lower risk of local inflammation following MSCs administration, reducing potential adverse effects [107]. In cattle, it has been demonstrated that fetal AT-MSCs and BM-MSCs responds to inflammatory stimulation with interferon γ (IFNγ) by increasing immune-related genes expression and activity in a dose-dependent manner, and up-regulating gene expression of IL-6 [30]. However, conditioned medium from IFNγ-stimulated and unstimulated BM-MSCs and AT-MSCs exert similar suppression of proliferation of alloantigen-activated bovine peripheral blood lymphocytes [30]. Whereas immunomodulatory properties appear to be similar between BM-MSCs and AT-MSCs, higher expression of MHC-I and MHC-II in BM-MSCs suggested that the immunogenicity potential of bovine fetal MSCs might be tissue-dependent and that AT-MSCs might be more suitable candidates for allogenic therapy [30].”

  1. The manuscript is missing the merits, and demerits of the use of cow MSCs in clinical field. 

“MSCs characteristics make them appealing for their potential in clinical applications and the lack of ethical concern is the other factor that makes them ideal for laboratory studies as well. As for humans [140], for successful cell-based therapies, stem cells must be able to differentiate into specific targeting cells, or must act via paracrine mechanisms. Their extraction and isolation must be feasible and transplantation must be effective and safe. Furthermore, a considerable number of cells is essential to optimize the therapeutic effects, requiring ex vivo cell expansion. However, the lifespan of MSCs is limited during in vitro culture and their senescence is a limit from the viewpoint of clinical applications. On one hand, the limited cell proliferation potency protects them from malignant transformation after transplantation, on the other senescence can alter various cell functions essentials for therapeutic efficacy, such as proliferation, differentiation, and migration. Therefore, after in vitro expansion and before therapeutic use, it should be considered whether these cells still possess stemness properties.”

  1. Other clinical applications as orthopedic applications were not highlighted. IF these applications were not applied clinically, this should be addressed in the conclusion and future perspectives section.

“MSC treatment has the potential to reduce animal recovery time and reduce economic loss associated with bone and joint injury, reducing the time for repair that can negatively influence milk and meat production and interfere with natural breeding [140]. Nevertheless, orthopedic applications has not yet been applied clinically in cows.”

  1. More focus should be given on the future directions and possible future applications of cow MSCs.

“Attitudes in the livestock industry have shifted towards the preservation of the commercial viability of individual animals with high genetic value, leading in turn to an increase in medical expenditure to keep those animals healthy [140]. MSC treatment has the potential to reduce animal recovery time and reduce economic loss associated with bone and joint injury, reducing the time for repair that can negatively influence milk and meat production and interfere with natural breeding [140]. Nevertheless, orthopedic applications has not yet been applied clinically in cows. The antimicrobial activity of MSCs and their derivatives has a great potential for the treatment of conditions such as mastitis. In addition to the direct impact on milk production in the dairy industry, it would provide an alternative to the use of antimicrobials reducing the possibility of antimicrobial resistance and the presence of antibiotics in milk. MSCs treatment has the potential to decrease the recovery from various diseases affecting production, thus increasing profitability.”

  1. The manuscript did not include any figures or illustrations that are important for the readers to understand the concept. 

Figures were added.

Round 2

Reviewer 1 Report

Although the information provided in the manuscript is more informative, the grammar used in the article makes it difficult to understand. The highlighted area lacks a clear idea, and although the changes made according to reviewers’ suggestions are correctly followed the terminologies used are not strongly suggesting what needed to be addressed. That’s why full proofreading is required before the final submission.

Author Response

We appreciate the reviewer's concern and we have submitted the manuscript for proofreading, aimed at an overall improvement of the text. We hope that the paper can now be accepted for publication.

Reviewer 3 Report

The authors have addressed the comments and the quality of the article improved. I recommend publication in the current form

Author Response

We thank the reviewer for the suggestions.